# Clinical Instructors’ Perspectives on the Assessment of Clinical Knowledge of Undergraduate Nursing Students: A Descriptive Phenomenological Approach

**DOI:** 10.3390/healthcare11131851

**Published:** 2023-06-26

**Authors:** Li-Chin Chen, Chun-Chih Lin, Chin-Yen Han, Ya-Ling Huang

**Affiliations:** 1Department of Nursing, New Taipei Municipal TuCheng Hospital (Built and Operated by Chang Gung Medical Foundation), New Taipei City 236, Taiwan; judy5612@cgmh.org.tw; 2School of Nursing, Chang Gung University of Science and Technology, Puzi City 61363, Taiwan; 3New Taipei Municipal TuCheng Hospital (Built and Operated by Chang Gung Medical Foundation), New Taipei City 236, Taiwan; cyhan@mail.cgust.edu.tw; 4School of Nursing, Chang Gung University of Science and Technology, Taoyuan City 33303, Taiwan; 5Faculty of Health (Nursing), Southern Cross University, Gold Coast, QLD 4225, Australia; ya-ling.huang@scu.edu.au; 6Visiting Research Fellow, Department of Emergency Medicine, Gold Coast Hospital and Health Service, Gold Coast, QLD 4215, Australia

**Keywords:** clinical instructor, undergraduate nursing students, clinical learning, descriptive phenomenological approach

## Abstract

This study aimed to determine clinical instructors’ perceptions of the assessments used to evaluate the clinical knowledge of undergraduate nursing students. This study uses a descriptive phenomenological approach. Purposive sampling was used to recruit sixteen clinical instructors for semi-structured interviews between August and December 2019. All interviews were audio recorded and transcribed verbatim. Data were analyzed using a modified Colaizzi’s seven-step method. Four criteria were used to ensure the study’s validity: credibility, transferability, dependability, and confirmability. Three themes were identified in the clinical instructors’ views on evaluating the clinical performance of student nurses: familiarity with students, patchwork clinical learning, and differing perceptions of the same scoring system. The study results suggest a need for a reliable, valid, and consistent approach to evaluating students’ clinical knowledge. If the use of patchwork clinical internships for student nurses is unavoidable, a method for assessing student nurses’ clinical performance that requires instructor consensus is necessary.

## 1. Introduction

Clinical learning is an essential component of nursing curricula, which is critical to nursing students’ development of professional competencies [1]. Clinical training ensures that graduates have the necessary knowledge, skills, attitudes, and clinical judgment to meet the needs of the population they will serve [2]. Nursing preceptors or mentors, referred to as clinical instructors (CIs), work closely with nursing students during clinical training to supervise their clinical training [3] and evaluate their clinical skills.

Evaluating student competence during clinical training is challenging for CIs [4,5], as they must assess students’ competence in technical procedures and ensure that students possess the theoretical knowledge, attitudes, and core values of nursing that will allow them to function safely and professionally [6]. Most clinical practice courses identify specific cognitive, affective, and skills-based learning goals [7]. However, these goals are abstract and difficult to use for assessing student competence as a learning outcome, as robust assessment criteria are needed.

While a systematic approach to clinical knowledge assessment is vital to student nurses’ continued learning [8], these assessments are often based on the CI’s perspectives, resulting in inconsistent evaluations and a lack of equity [9]. Furthermore, some CIs have been found to evaluate student clinical performance subjectively [10]. Although consistency in scoring among instructors has been shown to increase equity in the evaluation process and to ensure that all graduates have a pre-established level of clinical competence, the methods used to assess students’ clinical skills vary between CIs [6,8]. To identify factors that may affect the quality of the assessment system at one university nursing program, this study examines CIs’ experiences and perceptions of assessing nursing students’ clinical performance.

The university examined in this study has adopted a competency-based approach to clinical training and evaluation. Students are evaluated by CIs using a checklist that defines ten competency aims. These aims are weighted by a percentage based on their relative importance to clinical performance. However, empirical evidence has yet to be collected to measure the accuracy or validity of the checklist. It is also likely that all CIs across different divisions of the clinical setting have a different understanding of these ten competencies but use them identically. Moreover, various evaluation strategies are used to determine students’ final grades for clinical training, including quizzes, written reflection, written care plans, and actual care performance. Students’ clinical performance requires a reliable and valid method for standardized evaluations [4]. The information gathered from the CIs participating in this study may help implement any changes to improve the consistency and validity of students’ clinical performance assessments used in this university nursing program.

This study aimed to determine clinical instructors’ perceptions of using the nursing core values of clinical assessment provided by the university for students’ clinical performance.

## 2. Materials and Methods

### 2.1. Study Design

This study used Husserlian descriptive phenomenology to explore CI perceptions and experiences. This approach uses the concept of intentionality, which is a person’s directed awareness or consciousness of an object or event [11], to explore CI experiences with scoring students’ clinical performance. Husserlian phenomenology addresses how individuals experience phenomena; in other words, an individual’s consciousness of the world. According to Husserlian phenomenology, the effects of phenomena must be described precisely, and nothing should be added to these descriptions to prevent the inclusion of researchers’ interpretations. The present study focused on the lived experiences of CIs, which were described as phenomena to allow researchers to analyze the meaning of the CIs’ work experience and to understand their descriptions of their experiences. Moreover, the standards for reporting qualitative research (SRQR) checklist were followed in this study [12].

### 2.2. Participants and Setting

This study was conducted at a technical university in southern Taiwan. Nearly 1500 students are currently pursuing a bachelor’s degree in nursing, and 525 graduate each year from two- and four-year nursing programs. Graduates who finish their five-year junior nursing college program obtain a diploma in nursing and then enroll in the two-year baccalaureate nursing program. In contrast, high school graduates enroll in a four-year nursing program to approach their baccalaureate degree. Graduates from both two-year and four-year nursing programs obtain their bachelor’s degrees. However, students from the two-year program are licensed in nursing primarily, and students in the four-year program are pre-licensed. About 40 CIs are employed by the university to supervise the clinical training of these students. Purposive sampling was used to recruit participants who met the following inclusion criteria: (1) employed full-time as an instructor, (2) served as an instructor in a clinical field, and (3) had one year or more of experience working as a CI. Adjunct or substitute CIs and full-time nursing professors who occasionally guided student nurses in clinical practice were excluded.

### 2.3. Data Collection

Data collection began with an email sent to CIs to inform them about the nature and content of the study and to request voluntary participation. Next, the research assistant scheduled the times and dates of interviews with respondents willing to participate and who met the inclusion criteria. Data were collected from August to December 2019. The first author conducted these one-on-one, in-depth interviews in a quiet room at the university. A semi-structured interview format allowed the participants to describe their experiences freely. The interviewer requested clarification of participants’ accounts of their backgrounds or perspectives as necessary. A single interviewer conducted all interviews to ensure data collection consistency. Using an interview guide helped maintain focus and avoid bias toward the researcher’s areas of interest [13]. The interviewer shared a nursing background and native language with the participants, facilitating an in-depth understanding of their verbal and nonverbal cues.

The interviews were transcribed in Mandarin, audio-recorded, and transcribed verbatim by a co-investigator and an audio-typist who signed a confidentiality agreement. Corresponding transcripts and audio recordings were coded under the same filename. All interviews lasted 50 to 70 min, which enabled the collection of rich, detailed information and increased external validity. However, as the first author, the interviewer determined that sufficient data for the study had been collected after interviewing 16 participants; after that, no new information emerged. Data collection, therefore, ceased at that point.

### 2.4. Data Analysis

The data were analyzed using Colaizzi’s method [14] with a modified Colaizzi’s seven-step method [15]. First, all the transcripts were combined into one file for reading and initial data analysis to acquire a sense of the transcripts. Second, the first author and a co-investigator simultaneously listened to the audio recordings and read the transcripts line-by-line several times to understand the text’s overall meaning and extract significant statements. Third, the purposes of substantial statements were formulated and then organized into clusters of themes and sub-themes, which were then integrated into the descriptive phenomena and composed into descriptions to identify the phenomenon under study. Forth, the results were integrated into a detailed description of the phenomenon. The first author and a co-investigator with a similar background in qualitative research independently coded all transcripts. Fifth, consensus on the definitions was obtained through frequent discussions, and a detailed description of the study topic was formulated. Finally, member checking was performed on the transcripts and codes as a final validation of data analysis.

### 2.5. Ethical Considerations

This study was approved by the Institutional Review Board (201802251B0C601). All participants were fully informed about the study, including its purpose, the data collection procedures, potential risks and benefits, the time commitment required, and the protection of participants’ privacy and anonymity. Participants were advised that participation was entirely voluntary and that they had the right to withdraw from the study without penalty. No coercive or deceptive tactics were used to encourage participation.

### 2.6. Trustworthiness

Four criteria were used to guide and ensure the study’s trustworthiness: credibility, transferability, dependability, and confirmability [16]. Credibility was ensured via peer debriefing and member checks. In addition, all interviews were conducted by the first author (a qualified and experienced qualitative researcher with a doctoral degree and publications) to ensure the consistency and quality of the interview data. A study team member familiar with the research topic conducted a peer debriefing and reviewed the findings to ensure confirmability. Data accuracy was confirmed by each of the 16 participants, who received a copy of the transcript of their interview [17,18] and a summary of the codes [19]. All participants agreed that the codes reflected their perceptions and experiences with clinical training. The data collection and analysis methods have been described in detail to ensure dependability. The sample, inclusion, exclusion criteria, and recruitment process have been described in detail to ensure the study’s transferability.

## 3. Results

All participants were female instructors with extensive work experience in the Taiwanese healthcare environment before becoming CIs. The average work experience as a CI was 13.2 years, and most CIs held a master’s degree (81.3%). The demographic characteristics of participants are presented in Table 1. Three themes were identified as instructor perspectives on clinical learning by nursing students (see Table 2). All names in the Section 3 are pseudonyms.

### 3.1. Familiarity with Students

#### 3.1.1. Student Traits

Instructors indicated that their experiences with previous students may not apply to the current student cohort.


*“Previous students were obedient when confronting the challenges of clinical learning and humbly adopted instructors’ expectations. The current students tended to be self-centered and interested in expressing their views on learning but not hearing others’ views. However, they usually responded silently when they could not answer questions asked by CIs or preceptors.”*
(Lin)

The CIs tended to praise previous students and criticize current students, although they also recognized the strengths of current students and reported that past and present students had similar learning characteristics. The CIs perceived differences in their experiences with different students as student traits.


*“Previous students received substantial training in patient care, and students worked hard to ensure patient safety. The current students do what they are told and not more… The previous students were well-behaved and hardworking. Students today are self-centered and do what they think is best.”*
(Cheng)

#### 3.1.2. Learning Attitudes

The CIs compared the characteristics of their present students from different nursing courses.


*“Students in 4-year programs have better language skills and good logical thinking abilities. Although their performance in patient care techniques and communication with patients was unsatisfactory, these skills improved quickly. Students in 2-year programs performed care techniques well without thinking why to do it; they had strong intentions to pursue theoretical knowledge.”*
(Chen)

The CIs expressed concerns about students’ passive attitudes toward learning, suggesting that students were more concerned with their scores than what they learned. Students from different academic systems had different entry requirements, educational backgrounds, and clinical experience, resulting in various skill levels. However, the CIs emphasized the importance of teaching current students based on their aptitude. The scoring of the clinical performance of student nurses varied based on the CI’s perception of student attitudes. In other words, CIs used their subjective perceptions or feelings to evaluate student performance.


*“Students in the two-year program have good practical skills but a poor comprehensive understanding of professional knowledge. Students in the four-year program lack practical and professional knowledge in patient care.”*
(Wang)

CIs’ perceptions of student nurses’ learning attitudes led them to define what constitutes an active learner, demonstrating the CIs’ latent intention to train students to be active learners.

#### 3.1.3. Defining Active Learner

The CIs’ perspectives on the characteristics of students in different cohorts and programs functioned as evaluations that preceded students’ clinical performance scoring. CIs viewed students with higher scores as prepared to take on clinical care roles, while the opposite was true for those with lower scores. CIs also viewed active learners as serious students with a passion for learning, giving them higher scores. As Feng expressed:


*“Students with higher scores tend to be active learners. They take their work seriously, are proactive, academically sound, and demonstrate care skills. They also think critically and bring up problems. Students with lower scores cannot think critically and are passive learners.”*


#### 3.1.4. Demonstrating Care Behaviors

Instructors further associated higher scores with student “attitudes” and “mindsets,” but interviewees found it difficult to quantify these concepts. Academic knowledge and skills slightly influenced student scores and clinical performance. CIs admitted and scored soft power such as passion or mind of care, which they cared about. As Chen explained:


*Student nurses’ academic ability is not a significant factor in their clinical scores, but their mindset and initiative regarding learning play a more significant role. In addition, students who work with patients demonstrate their passion for interacting and communicating with them. However, such interaction and communication skills cannot be quantified.*


The CIs’ familiarity with the students thus is essential to their assessment of students’ clinical skills.

### 3.2. Patchwork Clinical Learning

#### 3.2.1. Incomplete Learning

Students’ clinical placement varied depending on the design of the curriculum. For example, a student enters the Adult Nursing practicum after completing Adult Nursing theoretical learning. This difference in student experience made it difficult to ensure consistency across learning experiences and that students internalized what they learned at each placement. Upon starting a new placement, some students had difficulty understanding the new material and often lacked the necessary background knowledge. As a result, instructors first needed to review what students should have learned during their previous clinical placement in a limited time. The instructors called this approach to clinical placements a “patchwork.” According to Shen:


*Student nurses’ internships are fragmentary and do not allow them to accumulate learning experiences. Each clinical placement is a new journey for student nurses. The instructors need to repeat the clinical content each time.*


#### 3.2.2. Lack of Focus on Clinical Learning

This patchwork approach to student clinical placements resulted from the brevity of the clinical learning journey and led CIs to question the ability of students to integrate care skills. Student nurses were unsure about the work needed to earn a good score. The tight clinical learning schedule led to high pressure and low motivation for learning. As a result, some students might disregard clinical training and focus on other things, such as their hobbies, instead. CIs asserted that the students were not serious about their internship work.


*“Student nurses are unable to manage their time during their studies effectively. They are not interested in nursing or people, nor do they have a feeling for illness and hands-on work, and they dislike practicing skills repeatedly. They are immersed in their interests, such as engaging in Cosplaying.”*
(Yang)

#### 3.2.3. Fulfilling the Learning Requirements

The CIs adopted specific strategies to meet the university requirements in light of the fragmented clinical internships. In one approach, CIs placed the responsibility for learning in the hands of the students. They provided positive feedback to encourage students’ clinical learning during their internships. Clinical training was central to students’ performance of care tasks, which reflected the CIs’ belief in learning by doing and scoring students’ attitudes toward learning. Zhu explains:


*“I need to comfort patients and their families, guide the nursing staff, and prepare students before going to the bedside and providing care. I always encourage, respect, and praise students for their learning achievements.”*


The CIs were warm and patient toward students, which facilitated better learning. In addition, they sometimes gave higher scores to increase students’ motivation and build their confidence. Xu explains:


*“I was impressed by an instructor who spoke warmly and kindly and explained things to students step by step. The instructor was very patient, and I learned from that instructor. A higher score can encourage students and increase their confidence.”*


### 3.3. Different Perceptions of the Same Scoring System

Although the university provides appraisal criteria based on the school’s ten values, CIs interpret and implement these criteria differently.

#### 3.3.1. Summative Evaluation

The instructors emphasized that the score for a clinical internship only represented a moment in time, which did not reflect the student’s past or future performance. They also highlighted that these scores indicated the instructor’s subjective assessment, not the student’s learning process. The subjective nature of these assessments might be another reason that CIs tended to give students higher scores for clinical training. A summative evaluation rarely emphasizes the student’s learning progress.


*“I will inform students that I am going to evaluate their clinical performance. I evaluate student performance during the last week of the clinical internship. I do not go back to evaluate the first week of the performance. Scoring depends on the teacher’s subjective view of the student nurse’s knowledge.”*
(Feng)

The CIs admitted that their evaluation of a student’s clinical internship is based on their subjective feelings. However, because the evaluation was made at one point, the CIs perceived that this evaluation was summative.

#### 3.3.2. Team-Based Evaluation

To reduce the subjectiveness of the clinical assessments, instructors sometimes scored the performance and aptitude of teams of students. First, an entire team of students was evaluated, and then team members were compared. This comparison provided a norm for intra-group evaluations of student knowledge. These evaluations tended to be more objective than individual evaluation alone. In addition, instructors perceived team-based assessments as fairer to the current students, as they were not based on instructors’ cumulative experience with previous students.


*“The evaluation involves comparing each student with six or seven other students in the group. This process defines high and low scores for the student group. Even scores for parameters such as attitude can be compared.”*
(Chao)

#### 3.3.3. Competence-Based Evaluation

The university provided competency-based evaluations (CBEs), which some instructors used to assess students’ clinical knowledge. Some instructors viewed the CBE as a complete kit for evaluating student knowledge in clinical settings. Others, however, questioned whether this tool could reflect or validate the professional knowledge and skills that underpin students’ care ability. However, the CBE included ‘soft’ skills such as diligence, respecting life, and accepting corrections humbly. Nevertheless, it was not easy to measure student knowledge effectively via CBEs, as students were still in the mindset of their training.


*“It is difficult to use the ten nursing core values to measure students’ caregiving abilities because the scores for professional knowledge and skills only account for fifty to sixty percent of the overall score. Thus, it is difficult to fail students even if they fail to take good care of patients from a professional standpoint.”*
(Lee)

The summative, team-based, and competence-based evaluations were eventually transformed into numbers to indicate a student’s clinical knowledge level.

## 4. Discussion

The CIs in the current study had contradictory views on “knowing students.” The CIs acknowledged the positive qualities of their current learners, including innovation, technological skills, and creativity; however, they did not believe these positive characteristics were essential to students’ professional development or clinical skills. Knowing a student’s strengths promotes fair student evaluation; thus, the CI’s familiarity with a student is essential to guarantee accurate student assessment [20]. Therefore, the CI’s attitude toward student strengths significantly influences students’ clinical learning and performance evaluations [21]. Furthermore, the CI’s perspective on students can determine whether the students are encouraged to be active learners [22]. When CIs unilaterally set all the learning goals and students passively accept these objectives, students become passive learners, simply doing what the instructor says or fulfilling the requirements. This situation may lead instructors to evaluate students as passive learners [9,23].

The relationship between the CI and students also affects the CI’s perceptions of a student’s clinical performance [17]. The CIs reported that the intermittent clinical placements of student nurses (patchwork clinical learning) made it impossible for students to experience continuous clinical learning and accumulate practical experience. The short duration of clinical placements also made it difficult for students to internalize what they learned. As a result, the content of previous clinical training sessions had to be repeated, and students could not connect the ideas on their own and move forward [24]. This situation is challenging to the professional development of both instructors and students. CIs must confront the pragmatic constraints of clinical patchwork learning to promote nursing students’ professional development [25]. However, this patchwork approach requires students to apply previously learned theoretical knowledge. Integrating theoretical knowledge and clinical experience is not the same as patchwork learning. Clinical learning is hindered when instructors view students’ clinical placements as patchwork learning because CIs view the students’ clinical experiences as less critical [26].

When student clinical internships are patchwork, instructors need to understand the students’ challenges and concerns and guide them through clinical training to ensure student satisfaction [27]. This guidance may significantly improve instructor-student relationships [28] and help bridge the differences between student and CI expectations of clinical placements. Unfortunately, because of this patchwork approach, some students have to repeat their clinical placements, which may increase their stress over developing their professional knowledge and skills [29].

The university provides CIs with a standard for evaluating student clinical exams. However, each CI interprets this standard differently, resulting in potential inconsistencies in CIs’ evaluations of student clinical performance. Consistent scoring of student performance indicates a CI’s objectivity and fairness [5]. As noted in the Section 3, different instructors used different methods (summative, team-based, and competence-based) to grade students’ clinical performance in various areas of professional practice (e.g., medical or orthopaedic). Regardless of the method used, the approach underlying all CI evaluations of students was subjective. CIs want to assess the clinical competence of prelicensure students accurately; therefore, a standard evaluation method is required to overcome this challenge [30].

A CI’s confidence significantly affects their assessment of students’ clinical performance [8]. Instructors must feel confident in their ability to grade students’ functional performance. McSharry and Lathlean [3] suggested that CIs need additional clinical teaching and learning education to prepare them to work as CIs. They also need considerable experience in guiding students to understand the standards of clinical grading fully. CIs may struggle to manage students’ learning and teach them how to deal with various clinical situations. It is essential for instructors to develop greater confidence in their knowledge and to develop better management strategies to guide student learning in the clinical setting [31].

### Study Limitations and Recommendations for Further Study

This study found that CIs viewed student nurses’ clinical internships as patchwork learning with inconsistent evaluation and demonstrated CI perceptions of the scoring system used to assess clinical internship performance. Although the data were collected at a single university, the small sample size may not reflect the perception of the majority of CIs. Large-scale multicentre studies are warranted to confirm the current findings. In addition, future research may benefit from quantitative analysis to complement CI evaluations of students in clinical learning.

## 5. Conclusions

The findings of this study provide valuable insights into CIs’ perspectives on assessing the clinical performance of student nurses. CIs perceived the clinical internships as patchwork because their time with each student was brief. The students often challenged their teaching strategies. CIs are essential to the learning model used at the university, and the learning environment created by a CI influences student outcomes in clinical training. If clinical internship assignments must be patchwork, it is crucial to establish a reliable, valid method for evaluating students’ clinical performance. CIs must agree on the tool used to grade clinical performance. More empirical studies in this area are needed to increase the objectivity of these evaluations and reduce the potentially negative impact of subjective judgments. Objective, fair, and transparent methods of evaluating students’ clinical performance are needed.

## Figures and Tables

**Table 1 healthcare-11-01851-t001:** Participant demographics.

	Degree	Sex	Age	Clinical Experience (Years)	Teaching Experience (Years)
1.	Master’s	F	53	4	24
2.	Master’s	F	54	7	23
3.	BSN	F	43	7	7
4.	Master’s	F	40	3	10
5.	Master’s	F	54	8	22
6.	Master’s	F	51	15	13
7.	Master’s	F	45	13	7
8.	Master’s	F	36	8	3
9.	Master’s	F	52	13	14
10.	BSN	F	44	4	17
11.	Master’s	F	54	20	10
12.	Master’s	F	39	7	7
13.	BSN	F	31	4	4
14.	Master’s	F	50	8	17
15.	Master’s	F	49	6	18
16.	Master’s	F	48	9	15

**Table 2 healthcare-11-01851-t002:** Themes and Subthemes.

Familiarity with students	Student traits
	Learning attitudes
	Defining active learner
	Demonstrating care behaviors
Patchwork Clinical learning	Incomplete learning
	Lack of focus on clinical learning
	Fulfilling the learning requirements
Different perceptions of the same scoring systems	Summative evaluation
	Team-based evaluation
	Competence-based evaluation

## Data Availability

Written informed consent was obtained from all subjects involved in the study.

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
