# Peer review of "Clinical Instructors’ Perspectives on the Assessment of Clinical Knowledge of Undergraduate Nursing Students: A Descriptive Phenomenological Approach"

_healthcare, 2023, doi:10.3390/healthcare11131851_

Round 1

Reviewer 1 Report

On p.3, lines 137-139, I think a citation on the technique of confirming data accuracy by allowing interviewees to confirm their responses would be helpful. There are some dissenting beliefs on that technique in research, so a citation would only strengthen the argument for using it.

Is the sample which this study was conducted on representative of all clinical instructors? The participant demographics are listed, but do these demographics generally match the population of clinical instructors. 

Throughout the results section, last names are still included. I believe it would be better to fully anonymize these statements.

P.8. The paragraph from line 329 to line 338 is confusing. This introduces that there is a standard evaluation for the university. If there is a standard of evaluation, the wrong question seems to be being asked. Everything in the manuscript points to needing reliable, valid, and consistent evaluations. This paragraph brings up that there is a standard. The question this study should then probably answer is why is that standard not being used. There's some talk in Section 3.3, but not much on how it is able to be interpreted in so many different ways. That is probably the most confusing part of this manuscript. Even the abstract decries the need for consistency in evaluation, but where a standard exists, it is not being implemented uniformly. The question seems to be more why is that and how do you improve that consistency.

p.2, Line 50. Looks like a key stroke error of "Cis" instead of "CIs"

p.2, Line 62. I'm not exactly sure what is being evaluated, the care plan or the number of care plans. If it is the number, "care plans written" is fine. But if it is just that care plans are evaluated, "written care plans" is clearer.

Author Response

Respond to the reviewer's comments as attached file. 

Reviewer 2 Report

Thank you for the opportunity to read the manuscript. Below are the most important notes:

- Introduction - line 54 - ...nursing school..maybe ...or university

- lack of purposefulness of research and information about what the work brings new

- Methodology: please describe the differences between the two nursing education systems (2-year and 4-year), please also specify your education system and nursing competencies (how many hours during your studies, in what mode, do you study in simulation centers, etc.) ) - this is important from the point of view of understanding the obtained results described in the Results part

Discussion - more references to world research; please refer to the  OSCE exam experience and your experience in the context of working with students in simulated conditions.

References - add several items; missing literature from 2022-23; correct the list of references - e.g. item 9 is quoted differently than the others.

Author Response

(The authors gave the same response as above.)

Reviewer 3 Report

Clinical training is very important in nursing education and is a serious problem. In this respect, evaluation of clinical ınstructors' perceptions may be important in solving problems. There are few studies on this subject. In this sense, the research topic contributes to the literature. In the introduction part, the deficiency in the literature should be emphasized more clearly. In the Method Section, it is stated that Colaizzi's seven-step method is used, but only four of them are explained. should be explained in other steps. In the findings section, examples of quotations should be included in Table 2. In general, this study was written in a very fluent and understandable manner in accordance with a qualitative study design. Thanks.

Author Response

(The authors gave the same response as above.)

Round 2

Reviewer 1 Report

Thank you for addressing the recommendations from reviewers. Very good job on this manuscript.